# Production of recombinant D-allulose 3-epimerase utilizing an auto-induction approach in fermentor cultures suitable for industrial application

Kenny Lischer[1], Fina Amreta Laksmi[2]*, Yudhi Nugraha[3], Helbert[4], Fauziah Az-Zahra[2], David Herawan[1,2], Ario Betha Juanssilfero[5], Des Saputro Wibowo[2], Kharisma Panji Ramadhan[2], Isa Nuryana[2], Mohd Shukuri Mohamad Ali[6]

**1** Department of Chemical Engineering, University of Indonesia, Jakarta, Indonesia, **2** Research Center for Applied Microbiology, National Research and Innovation Agency, Bogor, West Java, Indonesia, **3** Research Center for Molecular Biology Eijkman, National Research and Innovation Agency, Bogor, West Java, Indonesia, **4** Research Center for Ecology and Ethnobiology, National Research and Innovation Agency, Bogor, West Java, Indonesia, **5** Research Center for Environmental and Clean Technology, National Research and Innovation Agency, Bogor, West Java, Indonesia, **6** Enzyme and Microbial Technology Research Centre, Universiti Putra Malaysia, Selangor, Malaysia

* fina.amreta.laksmi@gmail.com, fina003@brin.go.id

## Abstract

D-Allulose 3-epimerase (DAEase) is the key enzyme catalyzing D-fructose to catalyze into D-allulose, a rare sugar in foods, which has lately drawn increasing worldwide attention owing to its possible health advantages and application as a substitute sucrose. This work focused on the development of an economical, scalable production method of DAEase by using the *Escherichia coli* BL21 star™ (DE3) as host expression. The research work aims to optimize the production of the enzyme through an auto-induction strategy in chemically defined media by using lactose as a natural inducer, thereby overcoming various limitations of conventional IPTG induction methods. The optimal concentration of lactose, glucose, and glycerol for maximum expression of DAEase was determined to be 1.5%, 0.125%, and 1.5%, respectively. Fermentor-scale optimization yielded a maximum amount of this enzyme with 5% inoculant, 300 rpm agitation, and 2 vvm aeration. Purification by affinity and anion exchange chromatography resulted in a sevenfold increase in specific activity with an overall yield of 12% and 43 mg of pure recombinant DAEase per liter of culture. Enzyme assays confirmed that DAEase had catalytic activity in the conversion of D-fructose to D-allulose, which was further confirmed by HPLC analysis. This optimized auto-induction-based strategy demonstrated its potential for large-scale production of DAEase in a cost-effective manner with enhanced reproducibility to meet industrial demands for synthesizing D-allulose.

**Data availability statement:** All relevant data are within the paper and its Supporting Information files.

**Funding:** This research was supported by the University of Indonesia (Grant No: NKB-508/UN2.RST/HKP.05.00/2024) in partnership with life and environmental research organizations, the national research and innovation agency (BRIN) (Grant No:1/III.5/HK/2024).

**Competing interests:** The authors have declared that no competing interests exist.

## 1. Introduction

D-Allulose is a rare sugar that received considerable attention worldwide because of its extremely low-calorie content and its effectiveness as an alternative for sucrose, providing 70% of the sweetness [1–3]. D-allulose may offer health advantages, especially in the context of diabetes and obesity management. The majority of D-allulose is taken up in the small intestine and eliminated from the body through urine. Furthermore, it is expected to be inactive regarding its metabolism of energy [4]. D-allulose has attracted significant interest worldwide as a low-calorie sweetener, recognized for its numerous potential health advantages. In 2014, it was granted Generally Recognized as Safe (GRAS) status by the U.S. Food and Drug Administration (FDA, GRN no. 498), permitting its application as a food ingredient and dietary supplement [3]. In Europe, D-allulose is classified as a novel food, necessitating approval from the European Food Safety Authority (EFSA) and the UK Food Safety Authority prior to commercialization [5]. The recent endorsement from the National Health Commission of China has significantly propelled the integration of D-allulose into a range of food and beverage products [2]. The increasing awareness among consumers regarding healthier sugar alternatives has led to notable expansion in the global market. The Insight Partners reports that the D-allulose market is anticipated to grow significantly, with projections indicating a rise to US$ 436.17 million by 2031, compared to US$ 263.06 million in 2023. The market is projected to achieve a compound annual growth rate (CAGR) of 6.5% between 2023 and 2031 [6]. In light of the growing demand, leading manufacturers have enhanced their production capabilities. Samyang Corporation, a prominent player in the food and chemical industry located in South Korea, operates a substantial D-allulose production facility. With an impressive annual production capacity of 13,000 tons, Samyang stands out as a significant player among industrial producers of D-allulose in the global market [7]. Tate & Lyle, a company based in the UK, stands out as a significant contributor to D-allulose production, driven by its continuous efforts to enhance production capacity and solidify its presence in the global market [8]. These advancements underscore the increasing market potential of D-allulose as a functional sweetener [2].

The bioproduction of D-allulose employing enzymatic conversion of D-fructose to D-allulose is favored due to its efficacy and environmentally friendly. The process converting D-fructose to D-allulose is initiated by the action of the enzyme, namely D-allulose 3-epimerase (DAEase), that modifies the C3 of D-fructose's hydroxyl group to produce D-allulose [9,10]. Since the discovery of the first enzymatic conversion by ketose 3-epimerase from *Pseudomonas cichorii* ST-24 in 1994 [11], DAEase from a variety of microorganisms have been found, each with unique biochemical characteristics [12–16]. The method utilized for DAEases expression was mainly based on traditional techniques, depending on Isopropyl ß-D-1-thiogalactopyranoside (IPTG) induction. Consequently, the reliance on the costly IPTG in conventional induction methods for expression strategies, together with the usage of intricate medium like LB, impedes its industrial applicability. Therefore, the advancement of more effective biosynthetic techniques via enzymatic pathway would not only diminish manufacturing expenses but also satisfy market demand and facilitate the extensive use of D-allulose.

This work seeks to provide an economically feasible approach for the industrial-scale synthesis of DAEase in *Escherichia coli* expression systems, emphasizing auto-induction, wherein lactose serves as a natural inducer instead of IPTG, with the utilization of a minimal defined media. To enhance the process, we adjust the composition of carbon sources and examine the impact of the supplemented carbon source on the production of recombinant DAEase. Research on the optimal conditions for DAEase production in a 5L fermentor was conducted as an initial inquiry before manufacture. The recombinant enzyme was purified to examine its activity.

## 2. Materials and methods

### 2.1. Materials

The pET28a-ApDAEase, developed by Laksmi et al. [17], encodes the gene for DAEase. The synthesize of DAEase was conducted utilizing *E. coli* BL21 star™ (DE3) (Invitrogen, USA) as the host organism. The recombinant DAEase was generated utilizing the pET-28(+) vector (Novagen, USA), which incorporates a T7 promoter and a kanamycin resistance gene for selection purposes. All compounds were of analytical quality and procured from commercial suppliers.

### 2.2. General expression of DAEase using induction technique

The expression induced by IPTG adhered strictly to the previously established conditions [17]. A recombinant colony was inoculated into 3 mL of Luria–Bertani (LB) broth supplemented with 30 ppm kanamycin, and the culture was grown at 37 °C with agitation at 165 rpm overnight. The subsequent day, the 1% preculture cells were transferred to a 125 mL shaker flask containing 30 mL of LB media. The flask culture was allowed to incubate at 37 °C until the cells achieved the exponential phase, which is indicated by an $OD_{600}$ of 0.8–1.0. Afterwards, 0.2 mM IPTG was added and incubated at 37 °C overnight to promote DAEase expression.

### 2.3. Auto-induction of DAEase in shake flask

Pre-culture (1%, v/v) was transferred into 30 mL of chemically defined media enriched with 30 ppm of kanamycin. The preparation of the chemically defined medium adhered to the methodology specified by Nugraha et al. [18]. The culture was then maintained under agitation at 165 rpm and a temperature of 37°C. The culture was subjected to centrifugation at 6000 xg for 6 minutes at 4°C to separate the cell pellet from the medium after an interval incubation time of 24 h, 48 h and 72 h. The process of ultrasonication for cell lysis entailed resuspending the cell pellet in a 50 mM Tris-HCl buffer at pH 8.0, followed by four cycles of sonication, with each cycle consisting of 30 seconds of sonication and 30 seconds of rest. The sample was kept on ice during the entire procedure to ensure a low temperature and avoid overheating, as it is established that maintaining cold conditions with interval cycles aids in preserving protein stability during sonication [19]. The crude recombinant DAEase was separated from cell debris through centrifugation (18,000 xg, 15 minutes, 4 °C). The expression of recombinant DAEase was evaluated on a 12% gel through SDS-PAGE, following the Laemmli method [20].

### 2.4. Carbon source concentration in chemically defined medium

The carbon source is crucial in shaping the composition of auto-induction media, as it directly influences the mechanism of protein expression in auto-induction conditions. This investigation concentrated on optimizing carbon sources, specifically glucose, lactose, and glycerol, while adhering to the media composition outlined by Nugraha et al [18]. The foundational medium exhibited a pH of 7.0 and was composed of a 1×phosphate/citric acid buffer, which included 1.7 g/L of citric acid, 13.3 g/L of $KH_2PO_4$, and 4 g/L of $(NH_4)_2HPO_4$, all dissolved in distilled water. The solution underwent sterilization in an Erlenmeyer flask at a temperature of 121 °C for a duration of 20 minutes. After reaching room temperature, 0.017 mM thiamine, 20 mM $MgSO_4$, 1×trace element solution, and the appropriate carbon sources were added under aseptic conditions. The 1×trace element solution was formulated from a concentrated 100×stock solution that includes 0.25 g/L of

CoCl$_2$·6H$_2$O, 10 g/L of Fe(III) citrate, 1.3 g/L of Zn(CH$_3$COO)$_2$·2H$_2$O, 0.15 g/L of CuCl$_2$·6H$_2$O, 0.25 g/L of Na$_2$MoO$_4$·2H$_2$O, 1.5 g/L of MnCl$_2$·4H$_2$O, 0.3 g/L of H$_3$BO$_3$ and 0.84 g/L of EDTA. The optimization of carbon supply (lactose, glucose, and glycerol) in the chemically defined medium was conducted to achieve the maximum yield of recombinant DAEase. For this purpose, the concentrations of lactose, glycerol, and glucose were adjusted, and the effects of each carbon source on the synthesize of recombinant DAEase were analyzed. Before optimization, the medium contained lactose, glycerol, and glucose at 0.5%, 0.9%, and 0.05%, respectively. The optimization process was systematically applied to attain the highest yield of recombinant DAEase, starting from lactose and then progressing to glycerol and glucose. The lactose concentrations were methodically adjusted to 0, 0.025, 0.05, 0.075, 0.1, 0.125 and 0.15%, while the concentrations of glycerol and glucose were fixed at 0.9%, and 0.05%, respectively. In the following effort to enhance the carbon supply, various glycerol concentrations of 0, 0.3, 0.6, 0.9, 1.2, 1.5 and 1.8% were incorporated into the medium. The lactose concentration that produced the optimal ratio of recombinant DAEase was utilized, while glucose was maintained at a constant level of 0.05% throughout the process. Additionally, the glucose concentration was adjusted to 0, 0.025, 0.05, 0.075, 0.1, 0.125 and 0.15% to optimize the yield of DAEase. At this point, the selected concentrations of lactose and glycerol were determined based on prior investigation. During every step of the optimization process, a qualitative analysis of the recombinant DAEase production was performed after a 24-hour incubation period. The measurement of soluble DAEase was conducted utilizing the gel analyzer software (USA). All experiments were performed in triplicate, and the results are expressed as the mean ± standard deviation to ensure reproducibility and statistical validity of the data.

## 2.5. Temperature variation for culture incubation

A further step involved adjusting the incubation temperature. The optimal concentration of the carbon source, which resulted in the highest DAEase yield in prior investigation, was utilized. The incubation temperatures were set at 18, 24, 30, and 37 °C. All experiments were performed in triplicate, and the results are expressed as the mean ± standard deviation to ensure reproducibility and statistical validity of the data.

## 2.6. Auto-induction of DAEase in a 5L fermentor

Following the attainment of the optimal carbon source composition for DAEase expression, it was subsequently utilized in a 5L fermentor. Subsequent efforts concentrated on refining the parameters of agitation, aeration, and inoculant concentration. A 5L New BrunswickTM BioFlo° 120 Bioreactor, featuring a Rushton blade (Eppendorf SE, Germany), was utilized for the fermentation process, operating at a working volume of 3 L.

A colony was introduced into 150 mL of LB medium, which was supplemented with 0.4% glucose and 30 parts per million (ppm) kanamycin, utilizing a 500 mL Erlenmeyer flask for the preculture. The culture underwent incubation at 37 °C with a rotational speed of 165 rpm, promoting its accelerated growth. The culture that had been incubated overnight was then introduced at a concentration of 5% (v/v) into a 3 L volume of LB broth that was supplemented with 30 ppm kanamycin (Sigma-Aldrich, USA). The experimental setup required the culture to be maintained at a temperature of 37 °C, with agitation levels observed between 200 and 400 rpm, while ensuring a consistent aeration rate of 2 vvm. The culture underwent incubation for a duration of 24 hours, with sampling conducted at 2-hour intervals throughout this period. After the centrifugation process, the cell was then suspended in a Tris-HCl buffer with a pH of 8 and a concentration of 50 mM. The recovery of crude protein from the suspension was achieved through the application of sonication. Following this, the crude protein was subjected to electrophoresis on a 10% acrylamide gel, and the analysis was performed using ImageJ (USA) software to quantify the yield of DAEase. After achieving the necessary level of agitation, the ideal aeration condition was then determined. The initial stages were executed employing the same methodology as the previous stages. The fermentation procedure was carried out at a temperature of 37 °C, utilizing the previously established agitation and maintaining aeration levels between 1–3 vvm. The cultures underwent a 24-hour incubation period. A total of 20 mL of the sample was systematically collected at 2-hour intervals for subsequent analysis. Following the established protocols,

each cell suspension received uniform treatment, and the crude protein was analyzed utilizing ImageJ software (USA). The last parameter that underwent optimization was the initial concentration of the inoculant. The preculture solution, with a concentration between 5% and 10%, was injected into the culture following the establishment of optimal aeration and agitation conditions. The culture was incubated at a temperature of 37 °C for a period of 24 hours. The methodology employed for sampling was consistent with that of the previous phase. An examination of the expression level of DAEase was performed utilizing Gel Analyzer (USA). All experiments were performed in triplicate, and the results are expressed as the mean ± standard deviation to ensure reproducibility and statistical validity of the data.

## 2.7. Growth rate assessment

The determination of colony forming units per mL (CFU/mL) was conducted through serial dilutions on LB agar that was enriched with 30 ppm kanamycin. The samples underwent incubation at a temperature of 37 °C for 16 hours prior to cell counting. Samples were cultured using LB agar plates in two replications. The maximal specific growth rate ($\mu_{max}$/h) was determined through linear regression analysis on plots illustrating the natural logarithm of the biomass (lnX) in relation to time (t).

## 2.8. Purification of DAEase

The purification of DAEase involved the use of a HisTrap HP column (Cytiva, USA) on an AKTA Prime Plus Liquid Chromatography System (Cytiva, USA). The column was equilibrated with buffer A, which consisted of 20 mM sodium phosphate, 500 mM NaCl, and a pH of 7.4, supplemented with 20 mM imidazole. Subsequently, the crude extract was automatically introduced into the column. Buffer B, consisting of 20 mM sodium phosphate, 500 mM NaCl, and adjusted to a pH of 7.4, was subjected to a linear gradient of 20–500 mM imidazole. This solution was passed through the column at a flow rate of 1 mL min$^{-1}$. The fractions beneath the peak were combined for additional purification using a HiTrap™ Q HP column (Cytiva, USA) on the same apparatus that had been equilibrated with buffer A (50 mM Tris-HCl buffer, pH 8.0). The elution of the protein was conducted with buffer B (50 mM Tris-HCl buffer, pH 8.0), employing a linear gradient from 0 to 500 mM NaCl, after which the target fractions were combined. The removal of NaCl was achieved through the dialysis of the pooled fractions using a 50 mM Tris-HCl buffer at pH 8.0.

## 2.9. Western Blotting

Western blotting was performed to verify the expression of recombinant DAEase, following the protocol outlined by the HisProbe™-HRP Conjugate kit (ThermoFisher, USA), with several modifications implemented. The purified protein underwent electrophoresis on a 10% acrylamide gel, followed by transfer to a nitrocellulose membrane using the Mini Protean® II trans blot unit (Bio-Rad). The recombinant DAEase was detected through the incorporation of the chromogenic substance 3,3′,5,5′-tetramethylbenzidine (TMB) onto the nitrocellulose membrane.

## 2.10. Determination of DAEase activity

Activity of DAEase was assessed by the quantifying D-allulose generated from D-fructose under controlled conditions. The assay was conducted in a 1 mL solution comprising D-fructose (100 mM), Tris-HCl buffer (50 mM, pH 8.5), 1 mM MgCl$_2$, and an enzyme concentration of 100 µg. The reactions occurred at a temperature of 70 ºC for a duration of 30 minutes, with the process being halted after 10 minutes through boiling. The enzyme unit is characterized by the total count of enzymes that facilitate the production of 1 µmol D-allulose per minute under optimal conditions, specifically at a temperature of 70 ºC and a pH of 8.5. The analysis of D-allulose is conducted using both quantitative and qualitative methods via HPLC (Shimadzu, Japan), which is outfitted with a refractive index detector and an HILICpak VG-50 4E column (Shodex). The temperature was maintained at 40 ºC, while deionized water was delivered at a flow rate of 1 mL min$^{-1}$. All

experiments were performed in triplicate, and the results are expressed as the mean ± standard deviation to ensure reproducibility and statistical validity of the data.

## 3. Results

### 3.1. DAEase expression using auto-induction method

The auto-induction method was selected to express DAEase. Transforming containing the pET28a-ApDAEase was grown for 24 hours at 37 °C in a chemically defined medium supplemented with 0.5% lactose to facilitate auto-induction. The DAEase employed in this study was noted to possess a molecular weight of approximately 36 kDa. For the expression assessment, *E. coli* BL21 star™ (DE3) lacking the pET28a-ApDAEase insertion served as a negative control. The negative control was grown under the same conditions as the recombinant cell. Following sonication, the supernatant of the sample and the negative control were subjected to SDS-PAGE analysis. The overexpression of recombinant DAEase in a soluble form was achieved through the auto-induction in a chemically defined medium. This was confirmed by SDS-PAGE analysis, which displayed a prominent band at approximately 36 kDa. Conversely, there was no detection of an overexpressed protein of comparable size in the negative control. A comparison with the induction method as positive control [17] was conducted, confirming that DAEase was successfully expressed (Fig 1). The impact of incubation duration on the production of recombinant DAEase was examined as well. The study demonstrated that a 24-hour incubation period in shake flask culture was adequate to achieve the maximum DAEase yield.

### 3.2. Carbon source optimization

The investigation focused on determining the optimal levels of glycerol, glucose, and lactose, in the chemically defined medium to boost DAEase production. The yield of DAEase demonstrated a favorable enhancement with the elevation of glucose concentration from 0 to 0.125%, as depicted in the figure (Fig 2A). However, a rise more than 0.125% in glucose concentration resulted in a decrease in protein yield. Fig 2B illustrates how varying lactose concentrations affect the production of DAEase. The enhancement in DAEase yield was evidently affected by a lactose increase of up to 1.5%. Nonetheless, levels surpassing 1.5% had a negative impact on protein production. The impact of glycerol concentration on DAEase synthesis was examined (Fig 2C). Increases in glycerol concentration up to 1.5% showed improvements in

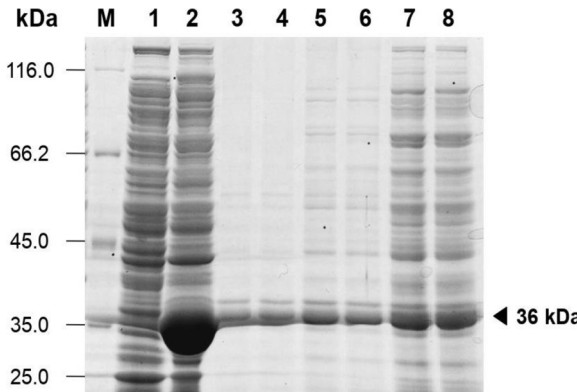

**Fig 1. Expression of DAEase using auto-induction. SDS-PAGE analysis of recombinant DAEase expressed in E. coli BL21 star™ (DE3) under auto-induction conditions.** The triangular sign indicated the specific protein band of interest. Lane M: protein marker Pierce™ Unstained Protein ladder, Thermo Fisher Scientific (USA); lane 1: negative control; lane 2: DAEase soluble fraction synthesized by induction using IPTG; lane 3 and 4: soluble fraction of DAEase produced using auto-induction method for 72 h incubation; lane 5 and 6: soluble fraction of DAEase produced using auto-induction method for 48 hours incubation; lane 7 and 8: soluble fraction of DAEase produced using auto-induction method for 24 hours incubation.

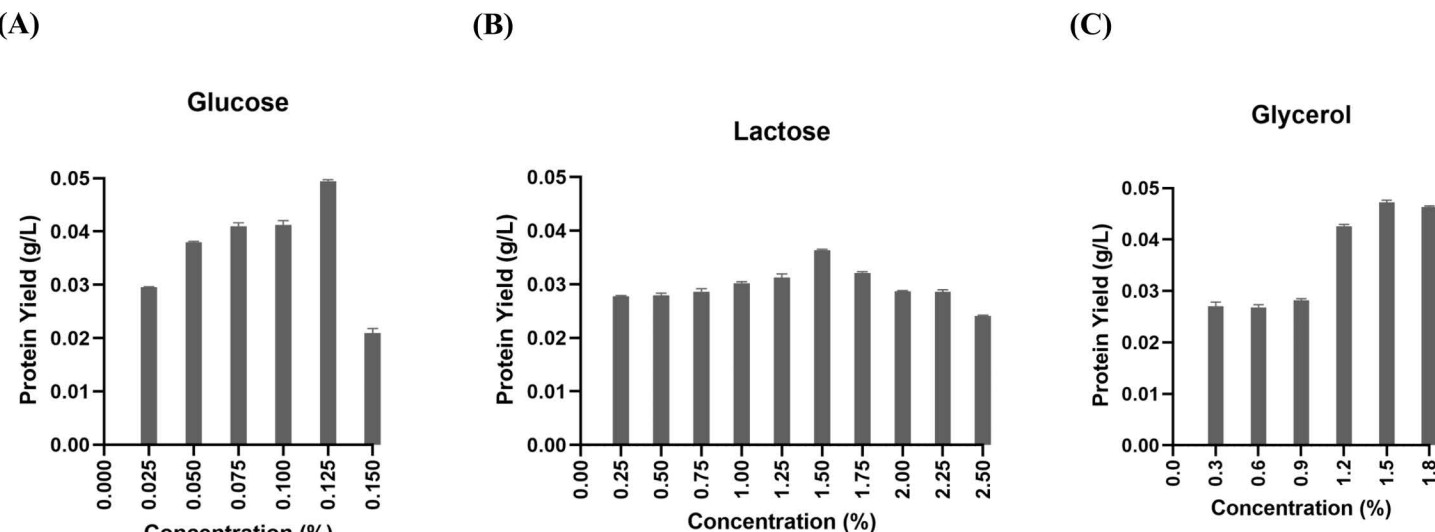

**Fig 2. Effect of carbon source for recombinant DAEase production using auto-induction method in chemically defined medium.** (A) The efect of glucose, (B) lactose, and (C) glycerol on the yield of Ap DAEase.

the yield of recombinant DAEase. However, the protein yields diminished at higher concentrations. Following this, we assessed the impact of reducing the induction temperature on DAEase expression (Fig 3). The findings indicated that temperatures below 37 °C were inadequate for the synthesis of DAEase, as they did not enhance the expression of the enzyme.

### 3.3. Effect of inoculant

After optimizing at the flask scale, the factors associated with fermentation conditions were subsequently assessed using a 5 L bioreactor. The first step in improving the production conditions of DAEase involved modifying the inoculant concentration while keeping aeration at 2 vvm with an agitation at 200 rpm. A transfer of 5% and 10% of inoculant was conducted to assess its impact on the culture (Fig 4A). The investigation revealed that the addition of 5%

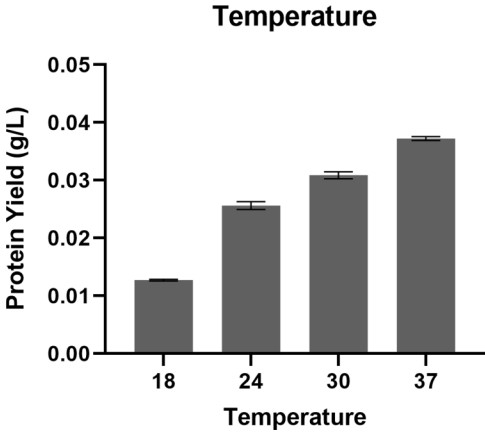

**Fig 3. Effect of temperature for recombinant DAEase production using auto-induction method in optimized chemically defined medium containing 1.5% lactose, 0.125% glucose and 1.5% glycerol at various temperature for 24 hours.**

**(A)**

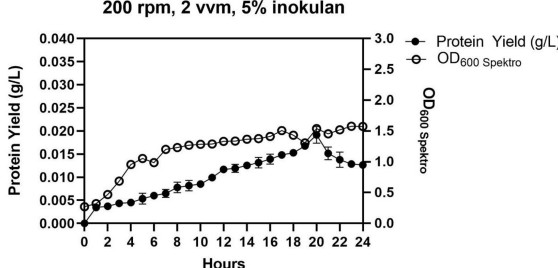

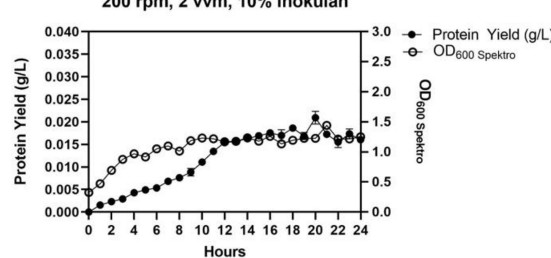

**(B)**

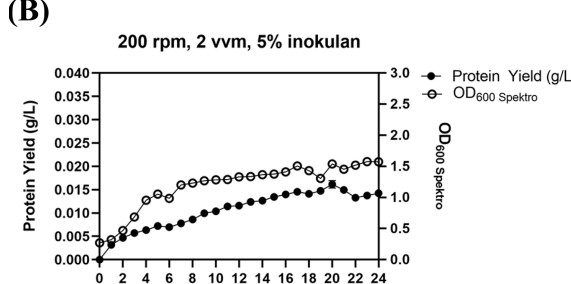

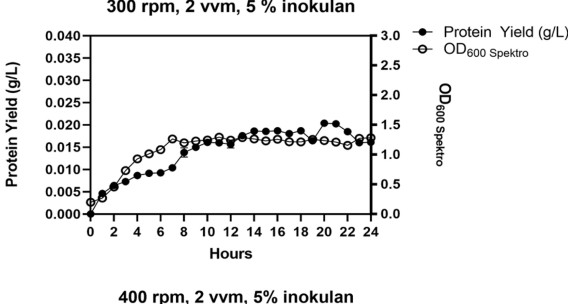

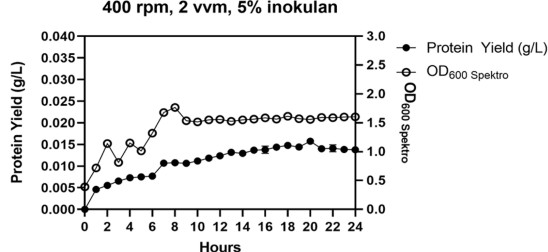

**(C)**

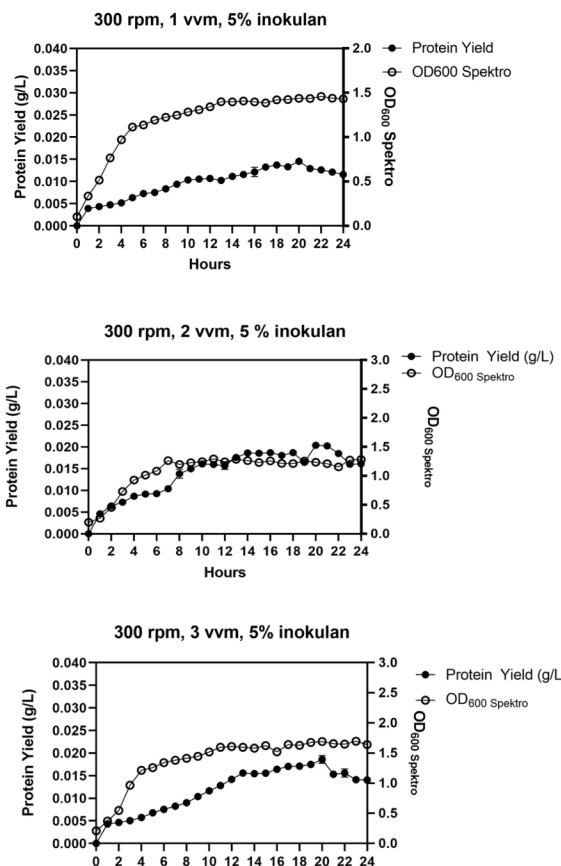

**Fig 4. Effect of aeration and agitation for recombinant DAEase production in 5L Bioreactor (working volume 3 L) using auto-induction method in optimized chemically defined medium containing 1.5% lactose, 0.125% glucose and 1.5% glycerol at 37 °C.** (A) Influence of inoculum concentrations at 5% and 10%. (B) Influence of agitation speeds at 200, 300, and 400 rpm during a 24-hour cultivation duration. (C) Influence of aeration rates at 1, 2, and 3 vvm during a 24-hour cultivation duration. Black dots (•) represent protein yield (g/L culture), while light dots (○) indicate $OD_{600}$ nm values.

inoculant resulted in greater DAEase production than the use of 10% inoculant. Furthermore, an increase in inoculant concentration was not attributed to an enhancement in microbial growth throughout the fermentation process. This finding suggests that a high cell density inoculum is not essential for achieving a positive effect on protein production and microbial growth.

### 3.4. Influence of agitation speed

The fermentations were conducted with a 5% inoculant and an aeration rate of 2 vvm, while varying the agitation speeds at 100, 200, and 300 rpm, respectively. Fig 4B demonstrates a noticeable increase in both DAEase production as agitation speeds were raised to 300 rpm, particularly evident after 20 hours of incubation, whereas the similar trend in all agitation conditions was observed in the microbial biomass. In contrast, increasing the agitation to 400 rpm resulted in a decrease in protein yield. As a result, it was concluded that 300 rpm represents the optimal condition for DAEase production.

### 3.5. Influence of aeration rate

The succeeding endeavor utilized several forms of aeration. Fig 4C illustrated that a positive trend was evident in DAEase production at aeration rates up to 2 vvm. Nonetheless, a comparable trend was not detected when the aeration was increased to 3 vvm. Despite microbial biomass exhibiting a similar trend to the previous aeration condition, a decrease in protein output was noted; hence, 2 vvm was chosen for further use.

### 3.6. Microbial growth rate

An examination of the microbial growth that occurred throughout the fermentation process was carried out, and the results are depicted in Fig 5. After a period of 6 hours of incubation, the exponential phase began. It was determined that after 10 hours of incubation, the maximum growth rate of about $2.9 \times 10^{10}$ CFU/mL was achieved, which corresponds to a maximum specific growth rate of 0.372 $\mu_{max}$/h.

### 3.7. Purification and activity of DAEase

The enzyme DAEase underwent purification to achieve electrophoretic homogeneity from crude extracts utilizing affinity chromatography and anion exchange chromatography techniques. The active fraction's purity was assessed through SDS-PAGE analysis (Fig 6A). Furthermore, western blot analysis was employed to validate the expression of the enzyme, utilizing the pure fraction of recombinant DAEase (Fig 6B). The analysis through western blot demonstrated that the auto-induction method successfully generated recombinant DAEase. The findings demonstrate that the recombinant DAEase was successfully expressed and purified, showing a protein band that aligns with earlier reports [17].

The assessment of the purification process of the enzyme can be conducted through the observation of its activity. The conversion from D-fructose to D-allulose was detected using HPLC to validate the enzymatic functionality of DAEase. Fig 7

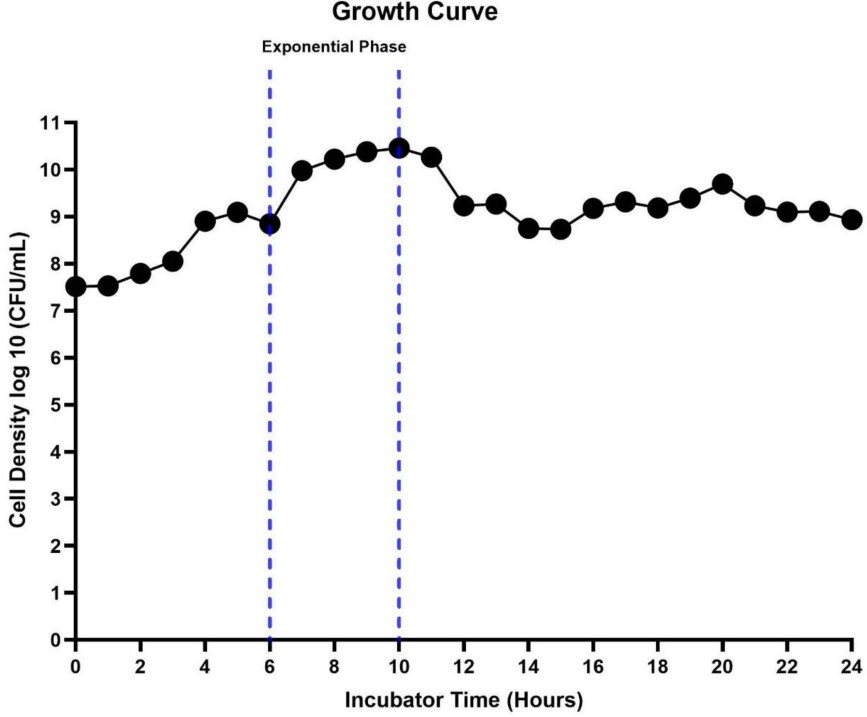

**Fig 5. Growth profile of recombinant DAEase in *E. coli* BL21 star™ (DE3) illustrating the temporal dynamics of cell growth.**

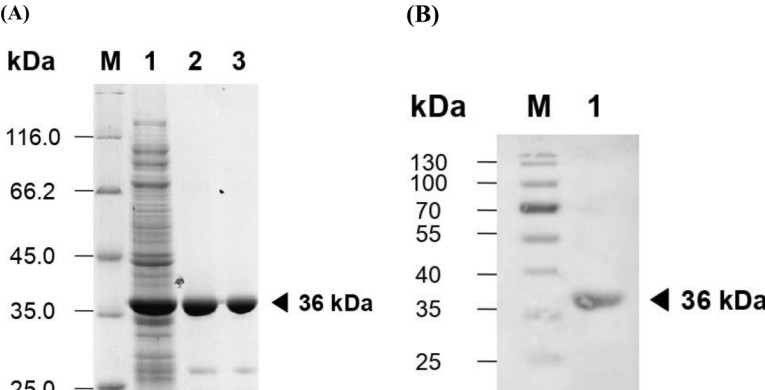

**Fig 6. Evaluation of pure DAEase through SDS-PAGE and western blot analysis.** (A) SDS-PAGE analysis of purifed DAEase. Lane M: marker; Lane 1: crude protein, Lane 2: HisTrap purifed DAEase; Lane 3: HiTrap purifed DAEase. (B) Western blot of DAEase. Lane M: marker; Lane 1: purifed fraction of DAEase.

presents the chromatogram of sugars D-allulose, which exhibits a retention time of 5.9 following the enzymatic reaction with DAEase. This result substantiates the effectiveness of DAEase in converting D-fructose to D-allulose. The information in Table 1 indicates that the rise in specific activity of the purified DAEase corresponds to an enhancement in the purification fold by as much as 7-fold, accompanied by an overall yield of 12%. From 1 liter of culture, a yield of 43 mg of pure recombinant DAEase was achieved.

## 4. Discussion

IPTG has several specific advantages, particularly in small-scale research, and is widely employed to induce expression from the lac promoter. The fact that the cell is unable to utilize IPTG makes it a metabolic-free inducer [21,22]. This keeps the induction level constant once IPTG is added to the growth medium. Nevertheless, IPTG has been identified as a possibly hazardous inducer, with evidence suggesting it may adversely impact cell growth [18,23–25]. The presence of these attributes constrains the use of IPTG for the protein synthesis at a larger scale. Auto-induction media represent a significant

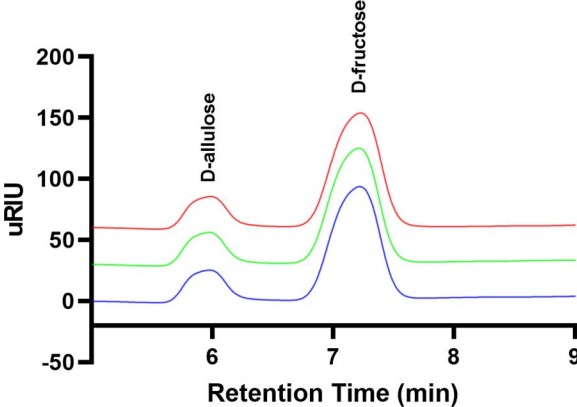

**Fig 7. HPLC chromatogram from the activity assay of DAEase produced using the optimized auto-induction method in chemically defined medium containing 1.5% lactose, 0.125% glucose and 1.5% glycerol at 37 ◦C.** The data represents triplicate results, depicted in different colors (blue, green, and red).

**Table 1. Purification step of recombinant DAEase produced using the optimized auto-induction method in chemically defined medium containing 1.5% lactose, 0.125% glucose and 1.5% glycerol at 37 ◦C.**

| Process | Total Activity (U) | Total Protein (mg per 1 L culture) | Specific Activity (U/mg) | Purification (fold) | Yield (%) |
|---|---|---|---|---|---|
| Crude Enzyme | 2149.674 | 1864.5 | 1.2 | 1.00 | 100 |
| HisTrap™ HP | 911.443 | 144.5 | 6.3 | 5.47 | 42 |
| HiTrap™ Q HP | 353.991 | 42.95 | 8.2 | 7.15 | 16 |

advancement in protein expression systems, eliminating the need for cell density tracking and the standard application of IPTG for induction [26]. The T7 promoter, present in pET vectors, allows for induction by both IPTG and lactose, the latter being a more cost-effective option, thereby enhancing the feasibility of large-scale protein synthesis [27–29]. The most recent reports indicate that this method has been used to synthesize recombinant proteins for a variety of purposes. In several instances, the overexpression of proteins including single-chain Fv (scFv) antibodies [30], antitumor bispecific fusion protein [31], lipase, reverse transcriptase [18], and Pfu DNA polymerase [32] was successfully achieved.

The lac operon-based auto-induction system is recognized as a highly reliable approach for protein expression, offering numerous benefits compared to the IPTG-induced expression system. This work thoroughly tackles prevalent issues linked to IPTG induction, such as incorrect protein folding, the formation of inclusion bodies, suboptimal protein expression yield, and metabolic stress. The media designed for auto-induction are rich in nutrients, facilitating bacterial growth and allowing for the spontaneous expression of recombinant proteins. The carbon sources present in the media are crucial to the auto-induction mechanism. Initially, glucose acts as the main carbon source for bacterial growth; once it is depleted, the cells transition to utilizing lactose. The transition involves the regioselective conversion of lactose into allolactose, which triggers transcription by liberating the repressor. Lactose is essential for the derepression of the lac promoter in the expression vector (pET28a), facilitating protein expression during auto-induction. Glycerol is incorporated to enhance the absorption of lactose after glucose levels have diminished. Furthermore, glycerol serves as an additional carbon source, contributing to the maintenance of a relatively stable pH during the cultivation process [33,34].

Complex medium and defined medium are two distinct types of media utilized for synthesis of recombinant protein. Luria Bertani (LB) medium stands out as a widely utilized complex medium for the expression of recombinant proteins. Nonetheless, the lack of a clearly defined composition presents various constraints, especially concerning large-scale or consistent production. The existence of unidentified amounts of peptides, amino acids, and various organic compounds can lead to variability between batches, complicating the effort to achieve consistent protein yields and quality [35–37]. Furthermore, the high nutritional content of LB medium can result in excessive growth and premature saturation of the culture, potentially disrupting the timing and effectiveness of induction. The identified factors render LB a less ideal choice for optimized and scalable expression systems, particularly when meticulous regulation of nutrient composition is crucial. Consequently, the application of defined media alongside lactose-based auto-induction presents a more reliable, manageable, and scalable option for the production of recombinant proteins [38,39]. The auto-inducing medium are designed to promote bacterial growth and the subsequent spontaneous expression of target proteins. They contain a combination of nutrients and carbon sources including lactose, and glucose, glycerol. After using glucose as a source of initial bacterial growth as well as repressor of protein expression, *E. coli* converts lactose to allolactose, which releases the repressor and starts transcription [26,29,40]. The inclusion of glycerol in auto-inducing media does not limit the depletion of glucose; rather, it facilitates the uptake of lactose once glucose is exhausted during bacterial growth. The pivotal function of the carbon source in facilitating the auto-induction of recombinant protein suggests that optimizing the carbon source included in the medium is likely to enhance protein yield. The auto-induction media typically comprises 0.2% lactose, 0.05% glucose, and 0.5% glycerol [26]; however, the concentrations may differ depending on the particular protein being produced [18,41–43]. The analysis of the effect of carbon source on protein yield in this investigation was conducted employing the

One-Factor-at-a-Time (OFAT) approach. This method assesses the impact of one specific variable on the outcome, maintaining all other variables at a constant level. The OFAT method offers a significant benefit by enabling precise observation of the distinct effects of each parameter. For instance, it facilitates the evaluation of how various factors affect protein yield during the process of recombinant protein expression. This approach is frequently employed to enhance expression conditions, including temperature, carbon source, aeration, agitation, and inoculum concentration, as documented by [44–46].

The optimization of carbon sources, including lactose, glucose, and glycerol, in a chemically defined medium was performed to attain the highest yield of recombinant DAEase. The initial concentration ranges for these carbon sources were derived from the research conducted by Studier et al [26]. In considering findings from Blommel et al [42], Sarduy et al [47], and Goudarzi et al [48], which involved the expression of luciferase, falcipain-2, and amylase, respectively, which demonstrated variability in the optimal range, we broadened the concentration ranges initially proposed by Studier et al [26]. This adjustment aims to enhance the precision of recombinant DAEase production optimization. After conducting optimization, the ideal amounts for lactose, glucose and glycerol were determined to be 1.5%, 0.125%, and 1.5%, respectively, resulting in the highest DAEase yield. The reduction in DAEase production at glucose concentrations exceeding 0.125% could be linked to catabolite repression and metabolic burden. In *E. coli*, elevated glucose concentrations can inhibit the activation of genes regulated by the lac operon, which includes those utilized for recombinant protein production. This phenomenon arises due to the preferential consumption of glucose over alternative carbon sources, which in turn postpones the uptake and metabolism of lactose, the inducer in auto-induction systems [27]. Furthermore, an overabundance of glucose can result in the buildup of inhibitory metabolic byproducts like acetate, which may hinder cell growth, diminish protein synthesis ability, and ultimately decrease the yield of the desired enzyme [49].

The process of fermentation demonstrates greater effectiveness in the bioreactor in comparison to the shake flask, primarily because the bioreactor allows for adjustments of certain process variables, thereby enhancing optimal microbial performance. While there are existing report on bioreactor-scale study concerning parameters for recombinant DAEase production using induction [50], there has been a notable limitation in scale-up studies utilizing both auto-induction method and chemically defined medium. Primary approaches in biotechnology for increasing yields and producing large quantities of recombinant protein with consistent quality include scaling up fermentation and optimizing processes [51]. The concentration of the inoculant stands as a pivotal factor influencing both of growth rate and protein synthesis in bioreactors. Studies indicate that the density of inoculant plays a crucial role in determining protein yields, with the ideal range varying based on the microorganism and protein involved [52,53]. In addition, aeration and agitation rates, together with all other known process factors and coefficients that have recognized physiological impacts, are appropriate to be used as physical scale-up parameters [54]. The shaking was optimized to guarantee a uniform distribution of the culture medium, ensuring an adequate supply of nutrients and oxygen [55,56]. Moreover, ideal shaking conditions inhibited cell aggregation, thereby enhancing bacterial growth and reproduction [57]. Meanwhile, aeration ensures a constant supply of oxygen in the culture medium, facilitates mixing, and eliminates gases generated during the process [56,58]. A constant supply of oxygen is essential in fermentation for the growth of aerobic microorganisms. Consequently, achieving precise control of agitation and aeration is crucial for optimizing bioreactor efficiency.

The cultivation occurred at a temperature of 37 °C, with agitation speeds varying between 200 and 400 rpm, while maintaining a constant aeration rate of 2 vvm. Agitation variation was implemented in accordance with earlier studies, which typically utilized 200, 300, and 400 rpm. Numerous studies indicated positive expression results when an aeration rate of 2 vvm was applied [59,60]. The selection of 37 °C is based on its representation of the optimal maximum growth temperature for *E. coli*. Initial experiments conducted at reduced temperatures of 18, 24, and 30 °C indicated that decreasing the temperature did not enhance the yield; rather, the optimal protein yield was achieved at 37 °C (Fig 3). One of the essential elements in the production of recombinant proteins is guaranteeing that they fold correctly into their functional three-dimensional structure. Although increased temperatures can occasionally result in incorrect folding and diminished solubility due to the rapid expression process [61], our study did not observe this phenomenon. Our findings

indicate that, even when produced at 37 °C, the protein yield was maximized and solubility was notably superior in comparison to the yields observed at 18 °C, 24 °C, and 30 °C. Furthermore, the functional activity assessments demonstrated that the enzyme expressed at 37 °C exhibited activity, suggesting that the protein not only achieved proper folding but also preserved its functional capabilities.

This study highlights the significant impact of agitation, aeration and inoculant concentration on the synthesis of DAEase in *E. coli*. The findings demonstrated that an inoculate concentration of approximately 5%, along with aeration at 2 vvm and agitation at 300 rpm, were sufficient to attain the highest yield of DAEase. Our investigation revealed that elevating agitation from 200 to 300 rpm resulted in a rise in protein yield, while a subsequent increase from 300 to 400 rpm led to a decline in protein yield (Fig 4B). The observed phenomenon can be linked to the shear forces produced at elevated agitation speeds, which have the potential to harm microbial cells. This damage may result in reduced growth rates, diminished product yields, and possible impairment of enzyme structures, as indicated by Iyer et al [60]. The process of aeration is vital as it improves the availability of oxygen and facilitates mass transfer, particularly concerning substrates and products. The findings indicate that enhancing aeration from 1 to 2 vvm led to a notable increase in protein yield, whereas a further rise in aeration from 2 to 3 vvm resulted in a minor reduction in protein yield (Fig 4C). This trend could be associated with the low solubility of oxygen in water, where excessive aeration might lead to turbulence, potentially diminishing the overall efficiency of oxygen transfer and placing stress on the cells. Efficient oxygen transfer in aerobic fermentation is crucial for optimal production, relying on both agitation and aeration [60]. Moreover, the concentration of inoculum influences the length of the lag phase, which is the time cells take to adjust before commencing active growth. The findings suggest that a 5% inoculum concentration was ideal for maximizing protein yield, with no notable difference observed when compared to the 10% concentration, as illustrated in (Fig 4A). This indicates that a moderate inoculum size achieves a balance between promoting growth and preventing swift nutrient depletion and oxygen limitation, which may arise at elevated inoculum concentrations, ultimately leading to a decrease in protein yield. On the other hand, a reduced inoculum size may prolong the lag phase, especially in nutrient-deficient environments, hindering growth and diminishing protein output [62].

The recombinant DAEase yielded 42.95 mg per liter culture after two-step purification using the auto-induction method. This is 2.4 times higher than the yield reported in the most recent study of DAEase production in a 5L bioreactor after one-step purification using an optimized induction method (17.6 mg per liter culture) [50]. Additionally, the auto-induction approach has already attained a fermentation length of 14–20 hours for maximal DAEase yield, which is equivalent to the induction method for manufacturing DAEase in a 5L bioreactor [50]. A prior study examined the expenses associated with producing recombinant reverse transcriptase (RT) in rich media using the auto-induction technique [63]. It has been observed that utilizing a lactose-based auto-induction system for protein expression is almost twice as cost-effective compared to the IPTG induction method, as evidenced by the production of reverse transcriptase (RT). Additional studies have shown that the auto-induction method can achieve cost efficiencies of four to five times in the production of serine hydroxymethyltransferase (SHMT) [64], while further research suggests it may be as much as 20 times more economical for expressing medium-chain fatty acids (MCFA) [65]. The results of this study indicate an approach that is both cost-effective and technically feasible for industrial production of DAEase.

## 5. Conclusion

Utilizing the auto-induction approach in a specified medium containing 0.125% glucose, 1.5% lactose, and 1.5% glycerol, about 42.95 mg of active DAEase per liter of culture was achieved. This was achieved under optimal conditions in a 5L bioreactor: 300 rpm, 2 vvm, and 5% inoculant. Findings gained from this work have crucial implications for future large-scale enzyme production, especially in the manufacture of D-allulose, a zero-calorie sweetener utilizing recombinant DAEase. The findings presented here are poised to highlight the economic viability of scaling up DAEase production through the auto-induction method in contrast to the traditional induction method for industrial applications.

## Supporting information

**S1 File. S1 raw images** Fig 1**. SDS-PAGE analysis of ApDAEase expression.** The supporting files consist of unprocessed raw images of Fig 1. SDS-PAGE analysis representing the expression of Ap DAEase. Images were captured using a mobile phone positioned above a white lightbox, without any modifications in brightness, contrast, or cropping. Lane M corresponds to the protein marker; lane 1 represents the negative control; lane 2 shows the soluble fraction of Ap DAEase produced via the IPTG induction method; lanes 3 and 4 depict the soluble fraction of Ap DAEase expressed using the auto-induction method with a 72-hour incubation; lanes 5 and 6 represent the soluble fraction produced with the auto-induction method after 48 hours of incubation; and lanes 7 and 8 illustrate the soluble fraction obtained through the auto-induction method following 24 hours of incubation. Lane X indicates an unused lane. **S2 raw images Fig 6A. SDS-PAGE analysis of purified ApDAEase**. The supporting files consist of unprocessed raw images of Fig 6A. SDS-PAGE analysis representing the purified of Ap DAEase. Images were captured using a mobile phone positioned above a white lightbox, without any adjustments in brightness, contrast, or cropping. Lane M corresponds to the protein marker; lane 1 contains crude protein; lane 2 shows the HisTrap-purified Ap DAEase; and lane 3 represents the HiTrap-purified Ap DAEase. Lane X indicates an unused lane. **S3.raw images** Fig 6B. **Western Blot of ApDAEase**. The supporting files consist of unprocessed raw images of Fig 6B. Western blot analysis illustrating the detection of Ap DAEase. Images were captured using a mobile phone positioned above a white lightbox, without any modifications in brightness, contrast, or cropping. Lane M corresponds to the protein marker, while lane 1 represents the purified fraction of Ap DAEase. Lane X indicates an unused lane. **S4 dataset Table 1. Purification Step of ApDAEase.**The supporting files comprise a dataset for Table 1 presenting the purification process of Ap DAEase, accompanied by supplementary tables that detail the step-by-step calculations. **S5 dataset Fig 2. Effect of Carbon Source (Glucose, Lactose, and Glycerol)**. The supporting files include a dataset corresponding to the graph presented in Fig 2, which illustrates the effect of concentration of different carbon sources—glucose, lactose, and glycerol—on protein yield. The dataset contains the data used for generating the graph, which was processed and visualized using GraphPad Prism. **S6 dataset Fig 3. Effect of Temperature**. The supporting files include a dataset corresponding to the graph presented in Fig 3, which illustrates the effect of temperature on protein yield. The dataset contains the data used for generating the graph, which was processed and visualized using GraphPad Prism. **S7 dataset Fig 4. Effect of Agitation, Aeration, and Inoculant Concentration**. The supporting files include a dataset corresponding to the graph presented in Fig 4, which illustrates the effect of agitation, aeration, and inoculant concentration on protein yield. The dataset contains the data used for generating the graph, which was processed and visualized using GraphPad Prism. **S8 dataset Fig 5. Colony Forming Units (CFU)**. The supporting files include a dataset corresponding to the graph presented in Fig 5, which illustrates the Colony Forming Units (CFU). The dataset contains the data used for generating the graph, which was processed and visualized using GraphPad Prism. **S9 dataset Fig 7. HPLC Chromatogram for ApDAEase activity assay.** The supporting files include a dataset corresponding to the graph presented in Fig 7, which illustrates the HPLC chromatogram for the Ap DAEase activity assay. The dataset consists of raw data obtained directly from HPLC analysis, containing retention time (minutes) and uRIU (micro refractive index units), which were subsequently processed and visualized using GraphPad Prism.
(ZIP)

## Acknowledgments

The authors acknowledge the support of the National Research and Innovation Agency of Indonesia (BRIN) Laboratory of Genomics for its facilities and technical assistance.

## Author contributions

**Conceptualization:** Fina Amreta Laksmi.

**Data curation:** Kenny Lischer, Fina Amreta Laksmi.

**Formal analysis:** Helbert Helbert, Kharisma Panji Ramadhan, Isa Nuryana.

**Investigation:** Fauziah Az-Zahra, David Herawan, Des Saputro Wibowo.

**Methodology:** Fina Amreta Laksmi, Helbert Helbert, Ario Betha Juanssilfero.

**Project administration:** Kenny Lischer, Yudhi Nugraha.

**Resources:** Kenny Lischer, Fina Amreta Laksmi.

**Supervision:** Fina Amreta Laksmi.

**Validation:** Kenny Lischer, Fina Amreta Laksmi, Yudhi Nugraha, Helbert Helbert, Mohd Shukuri Mohamad Ali.

**Visualization:** David Herawan.

**Writing – original draft:** Kenny Lischer, Fina Amreta Laksmi, Yudhi Nugraha.

**Writing – review & editing:** Kenny Lischer, Fina Amreta Laksmi, Yudhi Nugraha, Mohd Shukuri Mohamad Ali.

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
