## [Decision Letter · Decision Letter 0]

5 Mar 2025

Dear Dr. Laksmi,

We look forward to receiving your revised manuscript.

Kind regards,

Bashir Sajo Mienda, PhD

Academic Editor

PLOS ONE

Journal Requirements:

“University of Indonesia (Grant No: NKB-508/UN2.RST/HKP.05.00/2024) and life and environmental research organizations, the national research and innovation agency (BRIN) (Grant No: 1/III.5/HK/2024).”

6. Please include a separate caption for each figure in your manuscript.

7. Please include your tables as part of your main manuscript and remove the individual files. Please note that supplementary tables (should remain/ be uploaded) as separate "supporting information" files

Reviewers' comments:

Reviewer's Responses to Questions

**Comments to the Author**

1. Is the manuscript technically sound, and do the data support the conclusions?

Reviewer #1: Partly

Reviewer #2: Partly

2. Has the statistical analysis been performed appropriately and rigorously?

Reviewer #1: No

Reviewer #2: No

3. Have the authors made all data underlying the findings in their manuscript fully available?

Reviewer #1: Yes

Reviewer #2: Yes

4. Is the manuscript presented in an intelligible fashion and written in standard English?

Reviewer #1: Yes

Reviewer #2: Yes

Reviewer #1: Authors demonstrated the expression of DAEase enzyme by autoinduction in E. coli BL21 star to convert D-fructose to D-allulose. The strategy used for the expression of the enzyme is generally used in the lab, which is considered a conventional method without novelty.

1. In MM, authors failed to explain how or which mechanism used for autoinducing the expression of DAEase in which this may be a key for this manuscript.

2. CCD process with RSM methodology should be used to find the suitable levels of sugars used in defined medium and temperatures used for cultivation to maximize the expression instead of one-at-a-time method as authors performed.

3. This manuscript aimed to optimize condition for expressing DAEase then effects of carbon sources, temperature, aeration, agitation and incubation time should be plotted against the DAease activity instead of the protein yield. The protein yield did not directly reflect and correspond to the enzymatic activity.

4. L345-350; authors discussed the defined medium is suitable for using in the industrial scale. However, author did not mention the compositions of basic compositions of the defined medium while performed the optimization of carbon sources instead. Then, the composition of defined medium should be provided and its composition should be also optimized.

Reviewer #2: The manuscript presents a well-structured and comprehensive study on the expression, optimization, and production of recombinant DAEase using an auto-induction approach. The work is relevant to the field of enzyme biotechnology, particularly for industrial-scale applications such as D-allulose production. The authors successfully demonstrate the advantages of auto-induction over traditional IPTG-based induction, highlighting its cost-effectiveness, scalability, and efficiency in enzyme production. The experimental design is logical, and the findings contribute valuable insights into fermentation optimization. However, there are some points that require revisions before the manuscript can be accepted for publication.

Introduction

• Could the authors provide more context on the current market trends or quantitative data that illustrate the growing demand for D-allulose?

• The manuscript refered that conventional IPTG induction is costly and relies on complex media like LB. Can the authors elaborate on the specific limitations of these traditional methods, possibly with supporting data or literature comparisons?

• The authors chose an autoinduction strategy with lactose? How does this approach overcome the identified limitations in terms of cost-effectiveness and scalability?

Cultivation Conditions:

• Why was 0.2 mM IPTG chosen for induction? Were alternative concentrations or induction times evaluated to maximize DAEase expression?

• What is the rationale behind maintaining the culture at 37 °C with agitation levels between 200 and 400 rpm and a constant aeration of 2 vvm?

• Can the authors explain why a sequential approach (first varying lactose, then glycerol, and finally glucose) was chosen instead of a factorial design that could capture interactions between carbon sources?

• How were the specific concentration ranges (e.g., 0 to 0.15% for lactose) selected? Are these based on previous studies or preliminary experiments?

• Were any controls or validations performed to confirm that sonication efficiently lysed the cells without denaturing the enzyme?

• The enzyme activity assay is performed at 70 °C, pH 8.5, using a 100 mM D-fructose solution. How were these conditions selected as optimal for DAEase activity?

Results and discussion

• The study reports that increasing glucose concentration up to 0.125% enhances DAEase yield, but concentrations above this threshold reduce protein production. there an explanation (e.g., metabolic burden or repression) for this behavior?

• The data suggest that temperatures below 37 °C do not enhance DAEase expression. Did the authors evaluate both expression levels and enzyme activity at these different temperatures, and can they elaborate on how temperature may influence protein folding or solubility?

• The discussion states that agitation, aeration, and inoculant concentration are critical. Could the authors provide more insight into why higher inoculant concentrations (>5%) and more aggressive aeration/agitation negatively affected yields?

• Have the authors considered any alternative strategies (e.g., fed-batch approaches) that could potentially improve DAEase production even further?

• The discussion points out that auto-induction media can be significantly cheaper than IPTG-based methods. Can the authors include a more detailed breakdown of the costs (e.g., raw materials, labor, time savings) to quantify the economic benefit for large-scale DAEase production?

Other comments:

One key aspect missing from the discussion is a statistical analysis of the results. The study presents enzyme yield and optimization parameters but does not mention whether s

**Do you want your identity to be public for this peer review?** For information about this choice, including consent withdrawal, please see our Privacy Policy

Reviewer #1: **Yes: ** Prof. Kaemwich Jantama, Ph.D

Reviewer #2: No

---

## [Author Response · Author response to Decision Letter 1]

15 May 2025

Response to Reviewer

Manuscript Number : PONE-D-24-60329

Manuscript Title : Production of recombinant D-allulose 3-epimerase utilizing an auto-induction approach in fermentor cultures suitable for industrial application

Reviewer #1

1. In MM, authors failed to explain how or which mechanism was used for autoinducing the expression of DAEase in which this may be a key for this manuscript.

Response to Reviewer #1:

Thank you for your insightful comment. We acknowledge the importance of clarifying the mechanism behind the auto-induction of DAEase expression. In response to your suggestion, we have revised the Discussion section to include a detailed explanation of the lac operon based auto-induction system used in this study (Page17, Lines 382-396). The revised text explains the mechanism of the auto-induction system, including the role of carbon sources such as glucose, lactose, and glycerol in regulating the induction process.

2. CCD process with RSM methodology should be used to find the suitable levels of sugars used in defined medium and temperatures used for cultivation to maximize the expression instead of one-at-a-time method as authors performed.

Response to Reviewer #1:

Thank you for your insightful comment. In this study, we employed the One-Factor-at-a-Time (OFAT) method to optimize the conditions for protein expression. This approach was chosen because it enables a clear understanding of how each individual factor, such as carbon source and cultivation temperature affects protein yield, while keeping other variables constant. Our aim was to observe the direct influence of each variable to determine its contribution to protein production. This method has been commonly used for optimizing expression parameters in similar studies (Bandaipet et al., 2006; Zhou et al., 2018; Sumardee et al., 2020), and we have clarified this point in the revised manuscript (Page 19, Lines 420–428).

Reference:

Bandaiphet C, Prasertsan P. Effect of aeration and agitation rates and scale-up on oxygen transfer coefficient, kLa, in exopolysaccharide production from Enterobacter cloacae WD7. Carbohydr Polym. 2006;66(2):216–28. doi:10.1016/j.carbpol.2006.03.004

Zhou Y, Han LR, He HW, Sang B, Yu DL, Feng JT, Zhang X. Effects of agitation, aeration and temperature on production of a novel glycoprotein GP-1 by Streptomyces kanasenisi ZX01 and scale-up based on volumetric oxygen transfer coefficient. Molecules. 2018;23(1):125. doi:10.3390/molecules23010125

Sumardee NSJ, Mohd-Hairul AR, Mortan SH. Effect of inoculum size and glucose concentration for bacterial cellulose production by Lactobacillus acidophilus. IOP Conf Ser Mater Sci Eng. 2020;991:012054. doi:10.1088/1757-899X/991/1/012054

3. This manuscript aimed to optimize condition for expressing DAEase then effects of carbon sources, temperature, aeration, agitation and incubation time should be plotted against the DAease activity instead of the protein yield. The protein yield did not directly reflect and correspond to the enzymatic activity.

Response to Reviewer #1:

Thank you for your thoughtful comment. In this study, optimization was conducted to identify the optimal conditions for maximizing protein expression via auto-induction, with protein yield selected as the primary response variable to evaluate the effects of carbon sources, temperature, aeration, agitation, and incubation time. This approach aligns with previous studies, such as El-Baky et al. (2015), which also employed protein yield as an indicator of expression efficiency. Enzyme activity was subsequently measured using protein produced under these optimal conditions to confirm its functionality and ensure that the expressed enzyme retained its catalytic activity.

Reference:

El-Baky NA, Linjawi MH, Redwan EM. Auto-induction expression of human consensus interferon-alpha in Escherichia coli. BMC Biotechnol. 2015;15:14. doi:10.1186/s12896-015-0128-x

4. L345-350; authors discussed the defined medium is suitable for using in the industrial scale. However, author did not mention the compositions of basic compositions of the defined medium while performed the optimization of carbon sources instead. Then, the composition of defined medium should be provided and its composition should be also optimized.

Response to Reviewer #1:

Thank you for your insightful comment. In this study, our primary focus was on optimizing the carbon sources (glucose, lactose, glycerol), as they play the most direct and influential role in regulating protein expression under auto-induction conditions. The other components of the defined medium were described in the revised manuscript. This information has been added in the revised Materials and Methods section (Page 8, Lines 154-166).

Reviewer #2 (Introduction)

1. Could the authors provide more context on the current market trends or quantitative data that illustrate the growing demand for D-allulose?

Response to Reviewer #2:

Thank you very much for your insightful comment. We agree that providing quantitative data and contextual background on the current market trends of D-allulose helps strengthen the relevance of our study. In response, we have revised the manuscript to include detailed information on the regulatory status, market value, and production scale of D-allulose in the revised Introduction section (Page 5, Lines 75–95).

2. The manuscript refered that conventional IPTG induction is costly and relies on complex media like LB. Can the authors elaborate on the specific limitations of these traditional methods, possibly with supporting data or literature comparisons?

Response to Reviewer #2:

Thank you for your thoughtful comment. We appreciate the opportunity to clarify this point. In the revised manuscript, we have elaborated on the specific limitations associated with the use of LB medium in conventional IPTG-based expression systems. As a complex and undefined medium, LB introduces challenges such as batch to batch variability and a lack of control over nutrient composition, which can affect reproducibility and protein yield, particularly at larger production scales. These issues are supported by previous studies and have now been addressed in the revised Introduction section (Page 17, Line 398-409), with appropriate references added (Rosano et al., 2014; Terpe, 2006; Studier, 2005).

Reference:

Rosano GL, Ceccarelli EA. Recombinant protein expression in Escherichia coli: advances and challenges. Front Microbiol. 2014 Apr 17;5:172. doi:10.3389/fmicb.2014.00172.

Terpe K. Overview of bacterial expression systems for heterologous protein production: from molecular and biochemical fundamentals to commercial systems. Appl Microbiol Biotechnol. 2006;72(2):211–22. doi:10.1007/s00253-006-0465-8.

Studier FW. Protein production by auto-induction in high-density shaking cultures. Protein Expr Purif. 2005 May;41(1):207–34. doi:10.1016/j.pep.2005.01.016.

3. The authors chose an autoinduction strategy with lactose? How does this approach overcome the identified limitations in terms of cost-effectiveness and scalability?

Response to Reviewer #2:

Thank you for your valuable comment. In the revised manuscript, we have clarified the rationale for choosing a lactose-based auto-induction strategy and how this approach addresses limitations in cost effectiveness and scalability. The auto-induction with lactose process simplifies cultivation by including all required components at the start, eliminating the need for cell density monitoring or manual inducer addition. This makes the system more efficient and less labor intensive, which is advantageous for large scale production. It has also been reported that protein expression using a lactose based auto-induction system is almost twice as cost efficient as the IPTG induction method, as demonstrated in the production of reverse transcriptase (RT) (Handayani et al., 2024). Other studies have shown even greater cost advantages, the production of serine hydroxymethyltransferase (SHMT) using auto-induction was found to be four and five times more cost effective (Sopitthummakhun et al., 2012), while expression of medium-chain fatty acid (MCFA) showed that the method could be up to 20 times more affordable (Wang et al., 2009). These points have been added to the revised Introduction section (Page 22, Lines 517–523).

Reference:

Handayani CV, Laksmi FA, Andriani A, Nuryana I, Mubarik NR, Agustriana E, et al. Expression of soluble moloney murine leukemia virus-reverse transcriptase in Escherichia coli BL21 star (DE3) using autoinduction system. Mol Biol Rep. 2024;51:1–11. doi:10.1007/s11033-024-09583-6.

Sopitthummakhun K, Yuvaniyama J, Chitnumsub P, Kamchonwongpaisan S, Vanichtanankul J. Plasmodium serine hydroxymethyltransferase as a potential anti-malarial target: inhibition studies using improved methods for enzyme production and assay. Malar J. 2012;11:1–12. doi:10.1186/1475-2875-11-1.

Wang Z, Liu X, Zhang Y, Du G, Chen J. Engineering Escherichia coli for cost-effective production of medium-chain fatty acids from soy whey using an optimized galactose-based autoinduction system. Bioresour Technol. 2024;393:130145. doi:10.1016/j.biortech.2023.130145.

Reviewer #2 (Materials and Method)

1. Why was 0.2 mM IPTG chosen for induction? Were alternative concentrations or induction times evaluated to maximize DAEase expression?

Response to Reviewer #2:

Thank you for your thoughtful comment. In this study, the use of 0.2 mM IPTG for induction was intentionally chosen as it was successfully applied in a previously established expression protocol (Laksmi et al., 2022), which we followed without modification. Our aim was to maintain identical conditions in order to provide a reliable baseline for evaluating the effectiveness of the autoinduction strategy. The clarification has been added to the revised manuscript in the Methods section (Page 7, Line 129-130).

Reference:

Laksmi FA, Nirwanto R, Nuryana I, Agustriana E. Expression and characterization of thermostable D-allulose 3-epimerase from Arthrobacter psychrolactophilus (Ap DAEase) with potential catalytic activity for bioconversion of D-allulose from D-fructose. Int J Biol Macromol. 2022 Aug 1;214:426–38. doi:10.1016/j.ijbiomac.2022.06.117.

2. What is the rationale behind maintaining the culture at 37 °C with agitation levels between 200 and 400 rpm and a constant aeration of 2 vvm?

Response to Reviewer #2:

Thank you for your insightful comment. We have clarified the rationale behind the selected cultivation conditions in the revised manuscript (Page 20, Line 468–475). Specifically, 37 °C was chosen as it is the optimal maximum temperature for E. coli growth, and preliminary tests at lower temperatures (18, 24, and 30 °C) showed reduced protein yield. The variation in agitation speed (200-400 rpm) was based on previous literature, where similar ranges are commonly employed. Additionally, the aeration rate was maintained at 2 vvm, a condition frequently reported to yield favorable expression outcomes in the cultures (Bandaipet et al., 2006; Iyer et al., 2009; Abdella et al., 2020).

Reference:

Bandaiphet C, Prasertsan P. Effect of aeration and agitation rates and scale-up on oxygen transfer coefficient, kLa, in exopolysaccharide production from Enterobacter cloacae WD7. Carbohydr Polym. 2006;66(2):216–28. doi:10.1016/j.carbpol.2006.03.004

Iyer PV, Singhal RS. Glutaminase production using Zygosaccharomyces rouxii NRRL-Y 2547: Effect of aeration, agitation regimes and feeding strategies. Chem Eng Technol. 2009;32(12):1901–7. doi:10.1002/ceat.200900230.

Abdella A, Segato F, Wilkins MR. Optimization of process parameters and fermentation strategy for xylanase production in a stirred tank reactor using a mutant Aspergillus nidulans strain. Biotechnol Rep (Amst). 2020 Jun;26:e00457. doi:10.1016/j.btre.2020.e00457.

3. Can the authors explain why a sequential approach (first varying lactose, then glycerol, and finally glucose) was chosen instead of a factorial design that could capture interactions between carbon sources?

Response to Reviewer #2:

Thank you for your insightful comment. In this study, we utilized the One-Factor-at-a-Time (OFAT) approach to optimize protein expression conditions. This method was selected because it allows for a clear assessment of the effect of each individual factor, such as carbon source and cultivation temperature, on protein yield while maintaining other variables constant. Our objective was to specifically observe the impact of each parameter to understand its contribution to protein production. This strategy has been widely applied for optimizing expression conditions in previous studies (Bandaipet et al., 2006; Zhou et al., 2018; Sumardee et al., 2020), and we have clarified this aspect in the revised manuscript (Page 18, Lines 420–428).

Reference:

Bandaiphet C, Prasertsan P. Effect of aeration and agitation rates and scale-up on oxygen transfer coefficient, kLa, in exopolysaccharide production from Enterobacter cloacae WD7. Carbohydr Polym. 2006;66(2):216–28. doi:10.1016/j.carbpol.2006.03.004

Zhou Y, Han LR, He HW, Sang B, Yu DL, Feng JT, Zhang X. Effects of agitation, aeration and temperature on production of a novel glycoprotein GP-1 by Streptomyces kanasenisi ZX01 and scale-up based on volumetric oxygen transfer coefficient. Molecules. 2018;23(1):125. doi:10.3390/molecules23010125

Sumardee NSJ, Mohd-Hairul AR, Mortan SH. Effect of inoculum size and glucose concentration for bacterial cellulose production by Lactobacillus acidophilus. IOP Conf Ser Mater Sci Eng. 2020;991:012054. doi:10.1088/1757-899X/991/1/012054

4. How were the specific concentration ranges (e.g., 0 to 0.15% for lactose) selected? Are these based on previous studies or preliminary experiments?

Response to Reviewer #2:

Thank you for your valuable comment. We appreciate your suggestion and have provided a clearer explanation regarding the selection of concentration ranges for lactose, glucose, and glycerol in the revised manuscript (Page 19, Line 429-436). Initially, the concentration ranges were based on the study by Studier et al., 2005. However, as later studies by Blommel et al., 2007; Sarduy et al., 2012; and Goudarzi et al., 2016 showed that each protein may require different conditions, we expanded the ranges initially suggested by Studier et al., 2025 to ensure more accurate optimization of recombinant DAEase production.

Reference:

Studier FW. Protein production by auto-induction in high-density shaking cultures. Protein Expr Purif. 2005 May;41(1):207–34. doi:10.1016/j.pep.2005.01.016.

Blommel PG, Becker KJ, Duvnjak P, Fox BG. Enhanced bacterial protein expression during auto-induction obtained by alteration of Lac repressor dosage and medium composition. Biotechnol Prog. 2007;23(3):585–598. doi:10.1021/bp070011x.

Salas Sarduy E, Cabrera Muñoz A, Trejo SA, Chávez Planes MA. High-level expression of Falcipain-2 in Escherichia coli by codon optimization and auto-induction. Protein Expr Purif. 2012 May;83(1):59–69. doi:10.1016/j.pep.2012.03.008.

Goudarzi Z, Shojaosadati SA, Sajedi RH, Maghsoudi A. Optimization of auto-induction conditions for the heterologous expression of a maltogenic amylase in Escherichia coli. Appl Food Biotechnol. 2016 Mar 16;3(2):105–13. doi:10.22037/afb.v3i2.10484

5. Were any controls or validations performed to confirm that sonication efficiently lysed the cells without denaturing the enzyme?

Response to Reviewer #2:

Thank you for your insightful comment. In response to your suggestion, we have added a clarification in the revised manuscript to explain that the sonication procedure was carefully conducted under controlled conditions to minimize the risk of enzyme denaturation. Specifically, the sonication was performed in short intervals (30 s on and 30 s off) while the sample was kept on ice to maintain a cold environment and prevent overheating, which is critical for preserving enzyme activity (Saranya et al., 2014). This explanation has been added to the revised manuscript in the Methods section (Page 8, Line 144-149).

Reference:

Saranya N, Devi P, Nithiyanantham S, Jeyalaxmi R. Cells disruption by ultrasonication. BioNanoScience. 2014;4(4):335–7. doi:10.1007/s12668-014-0149-2

6. The enzyme activity assay is performed at 70 °C, pH 8.5, using a 100 mM D-fructose solution. Ho

---

## [Decision Letter · Decision Letter 1]

15 June 2025

Production of recombinant D-allulose 3-epimerase utilizing an auto-induction approach in fermentor cultures suitable for industrial application

PONE-D-24-60329R1

Dear Dr. Laksmi,

We’re pleased to inform you that your manuscript has been judged scientifically suitable for publication and will be formally accepted for publication once it meets all outstanding technical requirements.

Kind regards,

Bashir Sajo Mienda, PhD

Academic Editor

PLOS ONE

Additional Editor Comments (optional):

Reviewers' comments:

Reviewer's Responses to Questions

**Comments to the Author**

Reviewer #2: All comments have been addressed

2. Is the manuscript technically sound, and do the data support the conclusions?

Reviewer #2: Yes

3. Has the statistical analysis been performed appropriately and rigorously?

Reviewer #2: Yes

4. Have the authors made all data underlying the findings in their manuscript fully available?

Reviewer #2: Yes

5. Is the manuscript presented in an intelligible fashion and written in standard English?

Reviewer #2: Yes

Reviewer #2: The authors have carefully addressed all the comments. The corresponding changes have been made in the revised manuscript. This paper can be accepted

**Do you want your identity to be public for this peer review?** For information about this choice, including consent withdrawal, please see our Privacy Policy

Reviewer #2: No

---

## [Editor Report · Acceptance letter]

PONE-D-24-60329R1

PLOS ONE

Dear Dr. Laksmi,

I'm pleased to inform you that your manuscript has been deemed suitable for publication in PLOS ONE. Congratulations! Your manuscript is now being handed over to our production team.

Kind regards,

on behalf of

Dr. Bashir Sajo Mienda

Academic Editor

PLOS ONE